# An Integrated Research–Clinical BSL-2 Platform for a Live SARS-CoV-2 Neutralization Assay

**DOI:** 10.3390/v15091855

**Published:** 2023-08-31

**Authors:** Jing Zou, Chaitanya Kurhade, Hope C. Chang, Yanping Hu, Jose A. Meza, David Beaver, Ky Trinh, Joseph Omlid, Bassem Elghetany, Ragini Desai, Peter McCaffrey, Juan D. Garcia, Pei-Yong Shi, Ping Ren, Xuping Xie

**Affiliations:** 1Department of Biochemistry & Molecular Biology, University of Texas Medical Branch, Galveston, TX 77555, USA; jizou@utmb.edu (J.Z.); yanhu@utmb.edu (Y.H.); dabeaver@utmb.edu (D.B.); kytrinh@utmb.edu (K.T.); peshi@utmb.edu (P.-Y.S.); 2Department of Pathology, University of Texas Medical Branch, Galveston, TX 77555, USA; chaitanyakurhade28@gmail.com (C.K.); hcchang@utmb.edu (H.C.C.); jomeza@utmb.edu (J.A.M.); joseph.omlid@gmail.com (J.O.); baelghet@utmb.edu (B.E.); rgdesai@utmb.edu (R.D.); pemccaff@utmb.edu (P.M.); juandgar@utmb.edu (J.D.G.); 3Sealy Institute for Drug Discovery, University of Texas Medical Branch, Galveston, TX 77555, USA

**Keywords:** SARS-CoV-2, variants of concern, neutralization assay, COVID-19, live attenuated, high throughput, serological diagnosis

## Abstract

A reliable and efficient serological test is crucial for monitoring neutralizing antibodies against SARS-CoV-2 and its variants of concern (VOCs). Here, we present an integrated research–clinical platform for a live SARS-CoV-2 neutralization assay, utilizing highly attenuated SARS-CoV-2 (Δ3678_WA1-spike). This strain contains mutations in viral transcription regulation sequences and deletion in the open-reading-frames 3, 6, 7, and 8, allowing for safe handling in biosafety level 2 (BSL-2) laboratories. Building on this backbone, we constructed a genetically stable reporter virus (mGFP Δ3678_WA1-spike) by incorporating a modified green fluorescent protein sequence (mGFP). We also constructed mGFP Δ3678_BA.5-spike and mGFP Δ3678_XBB.1.5-spike by substituting the WA1 spike with variants BA.5 and XBB.1.5 spike, respectively. All three viruses exhibit robust fluorescent signals in infected cells and neutralization titers in an optimized fluorescence reduction neutralization assay that highly correlates with a conventional plaque reduction assay. Furthermore, we established that a streamlined robot-aided Bench-to-Clinics COVID-19 Neutralization Test workflow demonstrated remarkably sensitive, specific, reproducible, and accurate characteristics, allowing the assessment of neutralization titers against SARS-CoV-2 variants within 24 h after sample receiving. Overall, our innovative approach provides a valuable avenue for large-scale testing of clinical samples against SARS-CoV-2 and VOCs at BSL-2, supporting pandemic preparedness and response strategies.

## 1. Introduction

Severe acute respiratory syndrome coronavirus 2 (SARS-CoV-2) is the causative pathogen responsible for the coronavirus disease (COVID-19) pandemic. As of 26 July 2023, there have been more than 760 million confirmed cases of COVID-19, including 6.95 million reported deaths (https://covid19.who.int/(accessed on 26 July 2023)) worldwide. COVID-19 vaccines have effectively mitigated the pandemic and protected against severe illness and death. However, the emergence of new SARS-CoV-2 variants and the waning of vaccine-induced immunity over time have led to declined vaccine-mediated protection [1,2]. Neutralizing antibody response is one of the key parameters of immune protection. Therefore, monitoring neutralizing antibodies can assess vaccine effectiveness and immune evasion by new variants and provide insights on updating vaccine strategies.

While enzyme-linked immunoassays (ELISA) are commonly used for serological testing of IgM and IgG to determine the antibody status after infection or vaccination [3], they cannot detect neutralizing antibody levels. A neutralization test conducted under biological conditions remains the preferred and optimal method for evaluating the effectiveness of antibodies. A conventional plaque reduction neutralization test (PRNT) is considered the gold standard serological assay for measuring neutralizing antibodies [3]. However, PRNT is labor-intensive, difficult to automate, and time-consuming with low throughput, making it impractical for a large-scale and rapid analysis of clinical samples. Alternative methods, such as a live virus microneutralization assay [4,5,6] and fluorescence or luciferase reporter live virus neutralization assay [7,8,9], have been developed to address these challenges. Unfortunately, these assays require infectious SARS-CoV-2 in biosafety level 3 (BSL-3) facilities that are not widely available to all researchers and clinical laboratories. This limitation impedes COVID-19 diagnostics, surveillance, vaccine development, and therapeutic antibody testing. To overcome this hurdle, researchers have developed surrogate approaches for measuring neutralizing antibodies at BSL-1/2. These include an ELISA-based ACE2/RBD-binding inhibition assay [10], pseudovirus neutralization assay based on the backbone of the human immunodeficiency virus (HIV-1), murine leukemia virus, or vesicular stomatitis [11,12,13], as well as single-round infectious SARS-CoV-2 (SFV)-based neutralization test [14,15]. Among these approaches, the SFV-based neutralization assay stands out as SFV closely mimics the egress, maturation, and infection processes of authentic SARS-CoV-2 in biological cell conditions, while also being compatible with BSL-2 facilities. The SFV-based neutralization test has demonstrated high sensitivity, specificity, reproducibility, and accuracy [15]. Unfortunately, the production of SFV through the *trans*-complementation system is complex, challenging, and costly, potentially limiting its application in large-scale clinical sample testing. 

More recently, we developed a live-attenuated SARS-CoV-2 virus by modifying the viral transcription regulator sequence (TRS) and deleting the open-reading-frames (ORF) 3, 6, 7, and 8 (∆3678) [16]. The ∆3678 virus exhibits significant attenuation in primary human airway cultures, as well as in hamster and K18-hACE2 mouse models. Due to its substantial decrease in virulence, the virus has been conditionally approved by the National Institutes of Health (NIH) for use in BSL-2 laboratories. To facilitate high-throughput neutralization and antiviral testing, we engineered an mNeonGreen reporter ∆3678 virus (mNG ∆3678) [16]. Importantly, despite its delayed replication kinetics on VeroE6 cells, ∆3678 can propagate to a peak titer comparable to the authentic BSL-3 SARS-CoV-2 [16], providing an effective solution to the viral stock production bottleneck associated with SFV. Unfortunately, the mNG ∆3678-derived neutralization assay exhibited lower sensitivity than conventional PRNT [16], necessitating additional optimization. 

In this study, we generated a genetically stable mGFP ∆3678 with stronger fluorescent signals and sensitivity in a neutralization assay, compared to the original mNG ∆3678. Furthermore, we successfully constructed mGFP ∆3678 incorporating the spike gene from two predominant SARS-CoV-2 variants BA.5 and XBB.1.5. Subsequently, we established a comprehensive research–clinical BSL-2 platform by integrating the assay development laboratory, clinical sampling, automated neutralization assay, and patient reporting system. The performance of this platform was evaluated using human serum or plasma samples within clinical settings. 

## 2. Materials and Methods

### 2.1. Ethics Statement

The use of human COVID-19 sera was reviewed and approved by the Institutional Review Board (IRB#: 20-0070) at The University of Texas Medical Branch (UTMB). Patient consent was waived as the specimens were left over from the routine standard of care (SOC) collected retrospectively. No medical care decisions were based on the use of these specimens.

The use of Δ3678 SARS-CoV-2 at BSL-2 has been conditionally approved by the institutional biosafety committee at UTMB and the Office of Science Policy (OSP) from the National Institutes of Health (NIH). NIH approval is required for the use of Δ3678 SARS-CoV-2 in other laboratories. The automation systems in the UTMB neutralization lab were enclosed within a negative pressure containment (BioBUBBLE, Fort Collins, CO, USA). Personnel wore powered air-purifying respirators (Breathe Easy, 3M) with Tyvek suits, aprons, booties, and double gloves. The authentic virus used in this study was propagated and analyzed at the UTMB BSL-3 facility with redundant fans in the biosafety cabinets. 

### 2.2. Serum/Plasma Specimens

The leftover SOC clinical serum/plasma specimens were collected from individuals with or without COVID-19 infection or vaccination. They were categorized into four groups: (i) thirty human serum/plasma specimens collected before COVID-19 emergence (Appendix A); (ii) twenty human serum/plasma specimens collected in approximately 1 month after 2 doses of Pfizer-BioNTech COVID-19 mRNA vaccination (Appendix A); (iii) thirty human serum/plasma specimens collected from individuals with BA.5 infection and/or the bivalent booster (Appendix A); and (iv) twenty-five human serum/plasma samples collected from individuals after XBB.1.5 infection (Appendix A). All specimens were heat-inactivated at 56 °C for 30 min before testing. 

### 2.3. Cells

VeroE6 cells (ATCC CRL-1587) were maintained in a complete medium that contained high-glucose Dulbecco’s modified Eagle’s media (DMEM), 100 U/mL of penicillium–streptomycin (P/S), and 10% fetal bovine serum (FBS; Hyclone laboratories, South Logan, UT, USA). VeroE6-TMPRSS2 cells that expressed human transmembrane serine protease 2 (TMPRSS2) [17] were purchased from SEKISUI XenoTech, LLC, and grown in the complete media with an additional 500 µg/mL of Geneticin™ Selective Antibiotic (G418 Sulfate). A549-hACE2 cells were grown in the complete media with 10 µg/mL of blasticidin and 10 mM of HEPES at 37 °C with 5% CO_2_ [7]. All media, supplements, and antibiotics were purchased from ThermoFisher Scientific (Waltham, MA, USA). Cells were tested as mycoplasma negative.

### 2.4. Construction of mGFP Δ3678 and Omicron Subvariants BA.5 and XBB.1.5

The sequence-optimized mGFP (Appendix A) was cloned into the backbone of WA1 SARS-CoV-2 Δ3678 with a standard overlap polymerase chain reaction (PCR) using the same approach as described previously [16,18], resulting in mGFP Δ3678. The spike sequences of BA.5 (GISAID EPI_ISL_11542604) and XBB.1.5 (EPI_ISL_16292655) were introduced into the mGFP Δ3678 with PCR accordingly. The plasmid containing mGFP and spike variants was validated with Sanger sequencing. The full-length cDNA was assembled with in vitro ligation and purified as a template for in vitro transcription of the genomic RNAs using an mMESSAGE mMACHINE™ T7 Transcription Kit (ThermoFisher Scientific). The resulting full-length RNA and the N gene RNA transcripts were electroporated into the VeroE6-TMPRSS2 cells as described previously [18]. After 3 days of transfection, most cells exhibited green fluorescence and moderate cytopathic effects under a microscope. The supernatants (referred to as P0) were harvested. One milliliter of P0 was inoculated to freshly prepared VeroE6-TMPRSS2 cells for further amplification. After 2–3 days of infection, supernatants (P1) were clarified with centrifugation at 4 °C and 1000× *g* for 10 min, and frozen at −80 °C. Larger stocks (P2–P3) of mNG Δ3678_BA.5-spike and XBB.1.5-spike were amplified on VeroE6-TMPRSS2 cells following the same procedures as described above. The genome sequence of the passaged viruses was validated with Sanger sequencing or next-generation sequencing. 

### 2.5. Plaque Assay

Virion infectivity was quantified with a standard plaque assay using VeroE6-TMPRSS2 cells. Approximately 1.2 × 10^6^ VeroE6-TMPRSS2 cells were seeded to each well of six-well plates. The next day, 200 µL of 10-fold serially diluted viruses was added to the pre-seeded cells in 6-well plates. After incubation at 37 °C for 1 h, the inoculum was removed and 2 mL of overlay media that contained DMEM with 2% FBS, 1% penicillin/streptomycin, and 1% sea-plaque agarose (Lonza, Walkersville, MD, USA) were added to each well of 6-well plates. After a 3-day incubation, the plates were stained with neutral red (Sigma-Aldrich, St. Louis, MO, USA) and plaques were counted on a lightbox. 

### 2.6. RNA Extraction, RT-PCR, and cDNA Sequencing

Cell culture supernatants were mixed with a five-fold volume of TRIzol™ LS Reagent (ThermoFisher Scientific, Waltham, MA, USA) to extract the viral RNAs following the manufacturer’s instructions. The extracted RNAs were dissolved in 50 μL of nuclease-free water. For sequence validation of recovered viruses, 2 µL of RNA samples was used for reverse transcription by using the SuperScript™ IV First-Strand Synthesis System (ThermoFisher Scientific) with random hexamer primers. DNA fragments flanking the entire viral genome were amplified with PCR. The resulting DNA was cleaned up with a QIAquick PCR Purification Kit, and the genome sequences were determined with Sanger sequencing at GENEWIZ (South Plainfield, NJ, USA). Alternatively, the viral RNA samples were sent to the sequencing core at UTMB for next-generation sequencing. A pair of primers, F_primer (5′-tctctggcattaatgcttc-3′) and R_primer (5′-ctattggtgttaattggaacgcc-3′), was used in RT-PCR to amplify the cDNA between S and N genes in the viral genome to verify the presence of the mGFP gene. 

### 2.7. Negative-Staining Electron Microscopy

Virions were harvested from the supernatants of infected VeroE6 cells on day 3 post-infection and were pelleted down using a 20% sucrose cushion. The pellet was resuspended in 30 µL of a HEPES-saline buffer containing 10 mM of HEPES, pH 7.3, and 150 mM of NaCl. In total, 4 µL of the resuspended virions was applied to the CF200-Cu carbon film grids (EMS), which were cleaned in a Gatan 950 Solarus plasma cleaner. The grids were stained using 2% uranium acetate and imaged in a JEOL 2100FS microscope operated at 200 keV.

### 2.8. Fluorescent Focus Reduction Neutralization Test

A fluorescent focus reduction neutralization test (FFRNT) was performed on VeroE6 cells using mGFP Δ3678 according to an established protocol [9,16,19]. Briefly, VeroE6 cells were seeded in each well of black μCLEAR flat-bottom 96-well plates (Greiner Bio-one™) and incubated overnight at 37 °C with 5% CO_2_. On the next day, serum/plasma samples were 2-fold serially diluted in the culture medium with the first dilution of 1:20. The diluted sample was incubated with a hundred fluorescent focus units (FFU) of mNG Δ3678 at 37 °C for 1 h, after which the sample-virus mixtures were inoculated onto the pre-seeded Vero-E6 cell monolayer in 96-well plates. After 1 h of infection, the inoculum was removed and 100 μL of an overlay medium (DMEM supplemented with 0.8% methylcellulose, 2% FBS, and 1% P/S) was added to each well. The plates were then incubated at 37 °C for 16 h. Raw images of green fluorescent foci were acquired using Cytation^TM^ 7 (BioTek) armed with the 2.5 × FL Zeiss objective with a wide field of view and processed using the software settings (GFP [469,525] threshold 4000; object selection size 50–1000 µm). The foci in each well were counted and normalized to the non-serum-treated controls to calculate the relative infectivities. The curves of the relative infectivity versus the serum dilutions (log_10_ values) were plotted using Prism 10 (GraphPad). A nonlinear regression method with the log (inhibitor) vs. response-variable slope (four parameters) model (bottom and top parameters were constrained to 0 and 100, respectively) was used to determine the dilution fold that neutralized 50% of mNG SARS-CoV-2 (defined as FFRNT_50_) in Prism 10. 

### 2.9. Fluorescent Reduction Neutralization Assay

A fluorescent reduction neutralization test (FRNT) was established using A549-hACE2 cells. In total, 1.2 × 10^4^ cells in a 50 µL culture medium with 2% FBS was seeded in black µCLEAR flat-bottom 96-well plates (Greiner Bio-one). The next day, 2- or 3-fold serial diluted human serum/plasma was mixed with an equal volume of mGFP Δ3678 (100 fluorescent focus units) and incubated at 37 °C for 1 h. Afterward, 50 µL of virus–serum/plasma mixtures was added to each well of the black plates. After incubating at 37 °C and 5% CO_2_ for 16 h, 25 µL of a Hoechst 33,342 solution (400-fold diluted in Hank’s Balanced Salt Solution; ThermoFisher Scientific) was added to each well to counterstain the cell nuclei. After 20 min of staining, the fluorescent cells were scanned using the CellInsight CX5 high-content screening platform (ThermoFisher Scientific) with predefined threshold parameters obtained using non-infected and infected cells. The positive cells in each well were counted and normalized to the number of total cells, resulting in the infection rate. The infection rate from each well was finally normalized to the non-serum/plasma-treated controls to calculate the relative infectivities. The relative infectivity versus the serum/plasma dilutions (log_10_ values) was plotted using Prism 10. A nonlinear regression method with the log (inhibitor) vs. response-variable slope (four parameters) model (bottom and top parameters were constrained to 0 and 100, respectively) was used to determine the dilution fold that neutralized 50% of mGFP Δ3678 (defined as FRNT_50_) in Prism 10. 

### 2.10. Plaque Reduction Neutralization Test

The 50% plaque-reduction neutralization titer (PRNT_50_) was measured for each serum/plasma as previously reported [8]. Individual serum/plasma was 2-fold serially diluted in a culture medium with a starting dilution of 1:20. The diluted samples were incubated with 100 PFU of authentic USA-WA1/2020, BA.5, or XBB.1.5 spike variants. After 1 h of incubation at 37 °C, the serum/plasma–virus mixtures were inoculated onto 6-well plates with a monolayer of VeroE6 cells pre-seeded on the previous day. The PRNT_50_ value was defined as the minimal serum/plasma dilution that suppressed ≥50% of viral plaques. The neutralization titer was determined in duplicate assays, and the geometric mean was taken. 

### 2.11. Statistics

A correlation analysis was performed using the simple linear regression model in GraphPad Prism 10. The coefficient of variation (CV) is the ratio of the standard deviation to the mean. A measure of high-throughput assay quality, Z-factor (Z′), was calculated using the formula $Z^{\prime }=1-\frac{3\left(\delta_{P}+\delta_{n}\right)}{\left|\mu_{p}-\mu_{n}\right|}$. Means (µ) and standard deviations (σ) from both the positive and negative controls were calculated from several replicates.

## 3. Results

### 3.1. Characterization of mGFP Δ3678

The live-attenuated ∆3678 was constructed from the infectious clone of the strain USA-WA1/2020 SARS-CoV-2, which was first isolated in the United States in 2020 [20]. To facilitate high-throughput neutralization and antiviral testing, we incorporated a mutant TRS-driven mNeonGreen reporter downstream of the M gene in the ∆3678 SARS-CoV-2 backbone, resulting in mNG ∆3678 (Figure 1A). mNG Δ3678 has proven to be suitable for neutralization and antiviral testing [16]. To explore alternative fluorescent reporters, we replaced the mNG sequence with a modified green fluorescent protein (GFP) containing 76 synonymous mutations (Appendix A) to reduce potential homologous sequences within the GFP gene and mimic the codon usage pattern of authentic SARS-CoV-2 [21]. We assembled the infectious clone using the well-established in vitro ligation approach as described previously [18,20] and rescued the recombinant viruses by electroporating the in vitro transcribed genome-length RNAs of mGFP ∆3678 into VeroE6-TMPRSS2 cells. We recovered mGFP ∆3678 in the supernatants of electroporated cells (defined as P0) and passaged once on the VeroE6-TMPRSS2 cells to obtain the P1 virus. The P1 mGFP ∆3678 yielded infectivity with titers of 7 × 10^6^ PFU/mL, slightly higher than the mNG ∆3678. Moreover, P1 mGFP ∆3678 produced clear plaques with morphologies similar to the P1 mNG ∆3678 on VeroE6-TMPRSS2 cells (Figure 1B). Noticeably, WA1 SARS-CoV-2 exhibited higher titers and larger plaques when compared to both mGFP and mNG ∆3678 (Figure 1B), confirming the attenuation of Δ3678 in the cell culture. Sanger sequencing confirmed the absence of undesired mutations in the genome sequence of the recovered P1 mGFP Δ3678. Furthermore, a negative-staining electron microscopy analysis of P1 mNG Δ3678 virions revealed spherical particles with diameters of about 100 nanometers and characteristic crowns of peplomers (Figure 1C), similar to the authentic SARS-CoV-2 [22,23]. We observed consistent fluorescent signals through all seven passages of the mGFP Δ3678 on VeroE6-TMPRSS2 cells. RT-PCR results confirmed the retention of the mGFP gene in the passaged mGFP Δ3678 (Figure 1D). Next-generation sequencing of the P5 mGFP Δ3678 genome did not reveal any consensus changes, demonstrating the genetic stability of mGFP Δ3678 during cell culture propagation. 

A highly effective PRNT-equivalent fluorescence focus reduction assay (FFRNT) using SARS-CoV-2 mNG has been widely used to assess the neutralization of human serum against SARS-CoV-2 variants [9,19]. We previously demonstrated that mNG Δ3678 could also be used in FFRNT, although it was less sensitive than PRNT [16]. In this study, we sought to investigate the suitability of mGFP Δ3678 for FFRNT at BSL-2. Both mGFP and mNG Δ3678 exhibited green fluorescent foci on VeroE6 cells after 20 h of infection (Figure 1E). Interestingly, mGFP Δ3678 resulted in larger and brighter foci than mNG Δ3678 (Figure 1E and Appendix A). To evaluate the performance of mGFP Δ3678 in FFRNT, we tested 20 human serum samples collected from individuals who had received two doses of the Pfizer-BioNTech COVID-19 mRNA vaccine (Appendix A). mGFP Δ3678 FFRNT_50_ strongly correlated with PRNT_50_ using authentic SARS-CoV-2 (Figure 1F). Notably, mGFP Δ3678 (R^2^ = 0.78; Figure 1G) exhibited a higher correlation coefficient than mNG Δ3678 (R^2^ = 0.65) [16]. Moreover, the geometric mean ratio of FFRNT_50_ to PRNT_50_ for mGFP Δ3678 was higher than that for mNG Δ3678, indicating that mGFP Δ3678 has a higher sensitivity than mNG Δ3678 in FFRNT. Based on the superior characteristics of mGFP Δ3678, we focused on mGFP Δ3678 for the subsequent studies.

### 3.2. mGFP Δ3678-Based Fluorescence Reduction Neutralization Test (FRNT)

To improve the sensitivity and simplicity of the mGFP Δ3678-based neutralization assay, we developed an upgraded assay named the fluorescence reduction assay (FRNT) (Figure 2A). In this assay, we used A549-hACE2 cells instead of VeroE6 cells to enhance infection efficiencies, as A549-hACE2 cells ectopically express human angiotensin-converting enzyme 2. Furthermore, FRNT eliminated the step of removing the inoculum and adding a viscous culture medium after 1 hour of infection. Instead, dye for nucleus-counter staining was added as the last step for rapid image focusing and cell counting (Figure 2A). FRNT calculated the green fluorescent cells after 16 h of infection. The reduction in the infection rate (the ratio of green to blue fluorescent cells) reflects the inhibition with serum samples (Figure 2B). 

We calculated FRNT_50_ (a dilution that inhibits 50% of infection) from the neutralization curve (Figure 2C). A549-hACE2 cells were infected with mGFP Δ3678 at various MOIs to determine the optimal amount of the input virus. As expected, the infection rate proportionally increases with an increasing MOI (Figure 2D). A quantitative analysis revealed a linear correlation between the infection rate and MOI when the MOI was no more than 1.0; however, the correlation became nonlinear when MOI exceeded 1.0 (Figure 2D), suggesting that the input virus surpassed the available hACE2 receptors for infection. 

We chose MOI 0.5 for the FRNT to assess the assay specificity using 30 human serum/plasma samples collected before the COVID-19 outbreak (presumed negative). As expected, similar to PRNT using WA1 SARS-CoV-2 [8], there are no detectable neutralization antibody titers in these samples (Appendix A). Further, we compared the correlation and sensitivity of mGFP Δ3678-based FRNT with PRNT using the same set of 20 human serum/plasma samples described in Figure 1F,G. As expected, the neutralization titers generated with mGFP Δ3678-based FRNT showed a strong and significant correlation (R^2^ = 0.809) with PRNT using WA1 SARS-CoV-2 (Figure 2E). More importantly, mGFP Δ3678-based FRNT resulted in a geometric ratio of FRNT_50_/PRNT_50_ of 3.1 (Figure 2F), significantly higher than that obtained from the FFRNT (0.70 in Figure 1G). These results demonstrated that we successfully developed an optimized mGFP Δ3678-based neutralization assay with high specificity and sensitivity.

### 3.3. Integrated Bench-to-Clinics COVID-19 Neutralization Test Workflow

To facilitate rapid COVID-19 serological testing in clinical settings, we developed a Bench-to-Clinics COVID-19 Neutralization Test (B2CNT) workflow that involves multiple interconnected components (Figure 3A). The assay development lab is responsible for virus preparation and FRNT development. As part of the Clinical Laboratory Improvement Amendments (CLIA)-certified laboratories, a neutralization laboratory conducts assay validation and large-scale sample testing. To enhance efficiency and consistency, we automated the FRNT process at a neutralization lab by utilizing state-of-the-art equipment, such as robotic arms, liquid handlers, automated incubators, and high-content imagers. The robotic arms manage tasks such as sample transferring, barcode scanning, and tube decapping. Liquid handlers perform precise sample pipetting, serial dilution preparation, and liquid transferring during FRNT. Automated incubators maintain optimal cell culture conditions, ensuring reliable results. High-content imagers conduct image acquisition and a cellular analysis at the end of FRNT. A built-in scheduler coordinates the processes among various equipment. The neutralization titers are automatically reported on the patient’s medical charts upon passing quality controls. This workflow allows for accurate sample tracking through the system with minimal human intervention. The entire process was optimized to achieve a turnaround time of no more than 24 h after receiving samples.

We then assessed the effectiveness of the B2CNT workflow in responding to two predominant SARS-CoV-2 variants BA.5 and XBB.1.5. The assay development lab constructed mGFP Δ3678_BA.5-spike and mGFP Δ3678_XBB.1.5-spike by incorporating BA.5 and XBB.1.5 spike sequences (Appendix A) into the mGFP Δ3678 backbone, respectively (Figure 3A). Like mGFP Δ3678_WA1-spike, both mGFP Δ3678_BA.5-spike and mGFP Δ3678_XBB.1.5-spike exhibited robust fluorescence in infected A549-hACE2 cells (Appendix A). The optimal MOIs of these two variants for FRNT were determined to be 0.5–1, resulting in infection rates of approximately 20% (Figure 3B). Under these conditions (MOI, 0.5–1), high signal-to-noise ratios (>35) and Z′ factors (above 0.7) were also achieved across all the replicates (Figure 3B,C), indicating assay robustness and reliability. The neutralization lab further validated the mGFP Δ3678_BA.5-spike- or mGFP Δ3678_XBB.1.5-spike-based FRNTs by using the 30 human negative samples (Appendix A), 30 human serum/plasma specimens collected from individuals with BA.5 infection and/or the bivalent booster (Appendix A), and 25 human serum/plasma samples collected from individuals after XBB.1.5 infection (Appendix A). Both assays demonstrated 100% specificity in distinguishing positive (titer ≥ 20) and negative samples (titer < 20) (Appendix A). Additionally, strong correlations were observed between mGFP Δ3678_BA.5-spike (R^2^ = 0.9278) or mGFP Δ3678_XBB.1.5-spike (R^2^ = 0.7197) FRNTs and PRNT (Figure 3D–F). More importantly, the geometric ratios of FRNT_50_/PRNT_50_ were 2.4 for mGFP Δ3678_BA.5-spike and 3.3 for mGFP Δ3678_XBB.1.5-spike (Figure 3E–G), demonstrating their higher sensitivity than PRNT. 

To assess the reproducibility of the FRNT, two analysts performed mGFP Δ3678_BA.5-spike FRNT on four different days using the same set of 30 human specimens (listed in Appendix A), and one analyst performed the mGFP Δ3678_XBB.1.5-spike FRNT on three different days using the same set of 25 human serum/plasma samples (listed in Appendix A). mGFP Δ3678_BA.5-spike FRNT resulted in coefficient variations (CVs) of 9.27–30.26% (mean, 18.71%; median, 19.11%) across all 30 samples, while mGFP Δ3678_XBB.1.5-spike FRNT showed CVs of 5.55–45.2% (mean, 21.5%; median, 20.31%) across all 25 samples. There was 1 out of 25 samples tested in mGFP Δ3678_XBB.1.5-spike FRNT that showed a CV greater than 35%, a recommended limit for most cell-based assays [24,25], which may be due to poor sample quality. Overall, our results demonstrated that both mGFP Δ3678_BA.5-spike and mGFP Δ3678_XBB.1.5-spike FRNTs are highly sensitive, reproducible, and accurate. Therefore, we implemented the assays for COVID-19 testing in clinical settings.

## 4. Discussion

In this study, we successfully developed a second-generation BSL-2 live SARS-CoV-2 neutralization assay using the most recently developed live-attenuated SARS-CoV-2 (Δ3678) [16]. Leveraging this platform, we established the B2CNT workflow, which has proven to be highly efficient for rapid COVID-19 serological testing in clinical settings. 

The live-attenuated Δ3678 SARS-CoV-2 neutralization assay developed in this study offers several advantages. Firstly, similar to the single-round SARS-CoV-2 [14,15], the live-attenuated Δ3678 SARS-CoV-2 closely resembles the authentic SARS-CoV-2 in terms of its structural characteristics, maturation, and infection paths. This similarity allows the assay to generate neutralization results that closely approximate the gold standard PRNT. The mGFP Δ3678 FRNT exhibited comparable specificity and accuracy to PRNT using authentic SARS-CoV-2 in this study, making it a reliable alternative approach for measuring neutralization antibodies. Secondly, the mGFP Δ3678 replicates robustly and can reach high titers exceeding 10^6^ PFU/mL on VeroE6-TMPRSS2 cells. This high replication efficiency enables easy large-scale production of high-quality viral stocks, ensuring a sufficient supply and lesser cost for serological testing. Thirdly, the Δ3678 SARS-CoV-2 can be safely handled at BSL-2 laboratories, making it accessible to a broader range of researchers. This is a significant advantage, as BSL-2 facilities are more widely available and easier to work with than the higher containment BSL-3 laboratories. Furthermore, almost all clinical laboratories are at BSL-2. 

Fourthly, the assay has great versatility in analyzing SARS-CoV-2 variants. Using the well-established reverse genetic system, we can easily substitute the spike gene in the Δ3678 SARS-CoV-2 backbone with spike genes from other SARS-CoV-2 variants. This adaptability allows researchers to rapidly examine the neutralization evasion with SARS-CoV-2 variants at BSL-2, providing invaluable insights into the effectiveness of vaccines and therapeutics against newly emerged variants. 

Fifthly, this live-attenuated Δ3678 can be engineered to express various reporters beyond mNG and GFP, including luciferase. This versatility expands the utility of the assay for neutralization tests as well as antiviral screening. Notably, we showed the superiority of the sequence-optimized mGFP compared to mNG in the context of Δ3678 SARS-CoV-2, although mNG has proven to be brighter than GFP (human codon usage optimized GFP, a version that does not contain the 76 synonymous changes to match the viral genome) in vitro and in vivo [26,27]. Previously, we also observed stronger fluorescence signals produced by mNG but not the human codon usage optimized GFP in the context of authentic SARS-CoV-219. The underlying mechanism by which mGFP with codon usage-matched synonymous mutations improves the signals is not yet fully understood. We suspected that the reporter sequence might influence interactions among cis-elements in the SARS-CoV-2 genome. Our data strongly suggest that the proper design of reporter sequences may enhance assay robustness. 

mGFP Δ3678 can be used in the FFRNT (Figure 1F). However, mGFP Δ3678 FFRNT was less sensitive than PRNT (Figure 1G), potentially limiting its ability to detect samples with lower neutralization activities and resolve differences in neutralization against subvariants. Additionally, FFRNT involves complex and time-consuming steps, such as removing the inoculum and adding a viscous methylcellulose culture medium (see details in Materials and Methods) [9,19]. We addressed these limitations using more susceptible A549-hACE2 cells with optimized procedures, resulting in a modified version of the assay named FRNT. The FRNT has a reduced complexity as it does not require removing the inoculum or adding a viscous methylcellulose culture medium. Subsequently, we automated FRNT and incorporated it into a prototyped Bench-to-Clinics Neutralization Test (B2CNT) workflow. Using mGFP Δ3678 BA.5-spike and XBB.1.5-spike as examples, we demonstrated that B2CNT is highly specific, reliable, robust, and accurate in testing human serum/plasma samples against SARS-CoV-2 variants. Noticeably, the current B2CNT allows FRNT to be conducted in a 96-well plate format. With further refinements, such as the preservation of virus stocks at lower temperatures and the transition to a 384-well plate format, the capacity of B2CNT can be significantly elevated, while the assay cost can be substantially reduced. Collectively, the B2CNT workflow allows for a rapid response to the COVID-19 pandemic in clinical settings. 

In summary, the highly sensitive and robust fluorescent live-attenuated assay at BSL-2 laboratories in this study marks a significant advancement in COVID-19 serological testing. The B2CNT showcases the successful translation of research tools into practical clinical applications, offering a valuable tool for pandemic preparedness and infectious disease management.

## Figures and Tables

**Figure 1 viruses-15-01855-f001:**
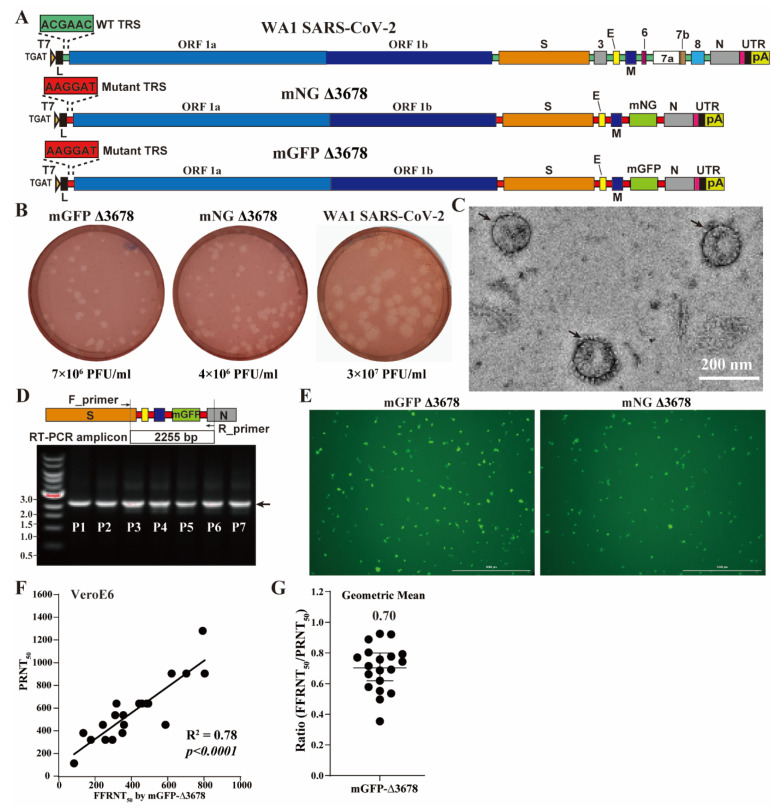
mGFP Δ3678 for neutralization assay. (**A**) Diagram of the construction of mGFP Δ3678. The wild-type and mutated TRSs are colored green and red, respectively. mNG, mNeonGreen; mGFP, sequence-optimized green fluorescence protein. (**B**) Plaque morphologies on VeroE6-TMPRSS2 cells. Numeric values indicate viral titers of P1 stock. (**C**) Representative negative-stained electron microscopy micrographs of mGFP Δ3678. Scale bar, 200 nanometers; arrows, virions. (**D**) Gel analysis. A replicon between F_primer and R_primer was amplified with RT-PCR from the P1-P7 passaged mGFP Δ3678 and analyzed on a 0.6% native agarose gel. DNA ladders are indicated. The arrow indicates the fragment containing the mGFP sequence. (**E**) Fluorescent focus on VeroE6 cells after 20 h of infection with mGFP Δ3678 or mNG Δ3678. (**F**) Correlation of mGFP Δ3678 FFRNT with PRNT. PRNT_50_ was obtained with plaque assay using strain USA-WA1/2020 at BSL-3. The coefficient and *p*-value (two-tailed) calculated from a linear regression model are shown. (**G**) Ratios of mGFP Δ3678 FFRNT_50_ to PRNT_50_. The geometric mean is shown. The error bar indicates the 95% confidence interval of the geometric mean. The FFRNTs using mGFP Δ3678 and mNG Δ3678 [16] were performed in parallel by using the same set of serum samples.

**Figure 2 viruses-15-01855-f002:**
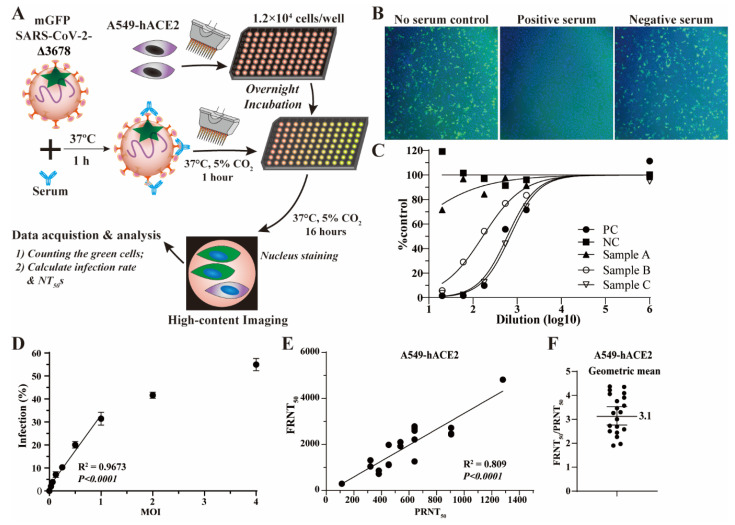
mGFP Δ3678-based FRNT. (**A**) Flow chart of mGFP Δ3678 FRNT on A549-hACE2 cells. (**B**) Representative fluorescent images after 16 h of infection. mGFP Δ3678-infected cells are shown in green. Blue indicates the nucleus. (**C**) Representative neutralization curve. PC, positive control; NC, negative control. (**D**) Infection rates at various multiplicities of infection (MOI). The coefficient and two-tailed *p*-value calculated from the linear regression model are shown. (**E**) Correlation of mGFP Δ3678 FRNT_50_ with PRNT_50_. PRNT_50_ was obtained with plaque assay using USA-WA1/2020. The coefficient and two-tailed *p*-value calculated from the linear regression model are shown. (**F**) Ratios of FFRNT_50_ to PRNT_50_. The geometric mean is shown. The error bar indicates the 95% confidence interval of the geometric mean.

**Figure 3 viruses-15-01855-f003:**
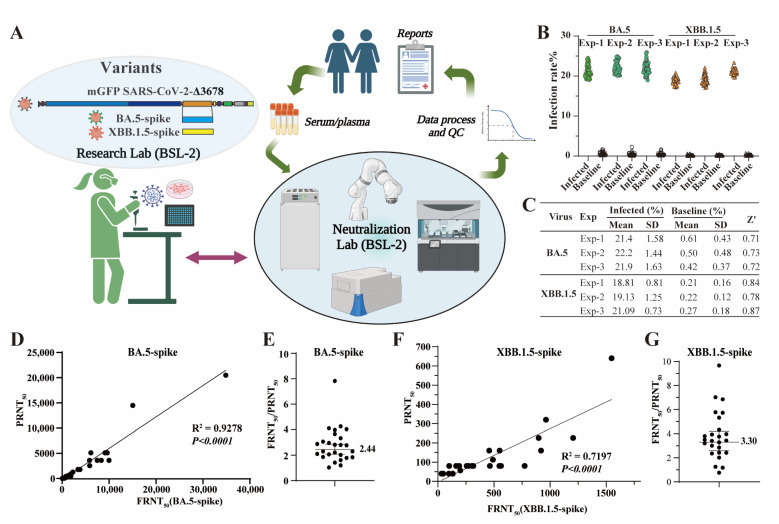
B2CNT for testing against SARS-CoV-2 variants BA.5 and XBB.1.5 at BSL-2. (**A**) Diagram of Bench-to-Clinics Neutralization Test (B2CNT) workflow. The schematic workflow was created with BioRender.com. (**B**) Infection rates after 16 h of infection with mGFP Δ3678_BA.5-spike or mGFP Δ3678_XBB.1.5-spike. (**C**) Analysis of the assay robustness. SD, standard deviations; Z’ > 0.5, suitable for high-throughput screening. (**D**) Correlation of mGFP Δ3678_BA.5-spike FRNT_50_ with PRNT_50_. PRNT_50_ was obtained with plaque assay using the USA-WA1/2020 containing BA.5-spike. The coefficient and *p*-values (two-tailed) calculated from a linear regression model are shown. (**E**) Ratios of FFRNT_50_ using mGFP Δ3678_BA.5-spike to PRNT_50_. The geometric mean is shown. The error bar indicates the 95% confidence interval of the geometric mean. (**F**) Correlation of mGFP Δ3678_XBB.1.5-spike FRNT_50_ with PRNT_50_. The coefficient and *p*-values (two-tailed) calculated from a linear regression model are shown. (**G**) Ratios of FFRNT_50_ of mGFP Δ3678_XBB.1.5-spike to PRNT_50_. The geometric mean is shown. The error bar indicates the 95% confidence interval of the geometric mean.

## Data Availability

The original contributions presented in the study are included in the article. Further inquiries can be directed to the corresponding authors.

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
