# Peer review of "An Integrated Research–Clinical BSL-2 Platform for a Live SARS-CoV-2 Neutralization Assay"

_viruses, 2023, doi:10.3390/v15091855_

Round 1
Reviewer 1 Report
- Major comments:
In this study, Jing Zou et al. presented an integrated research clinical platform for live SARS-CoV-2 neutralization assay, utilizing a highly attenuated SARS-CoV-2 (mGFP Δ3678_WA1/BA.5/XBB.1.5-spike). The viruses exhibited robust fluorescent signals in infected cells, and neutralization titers in an optimized fluorescence reduction neutralization assay that highly correlates with the conventional plaque reduction assay.
Furthermore, they established a streamlined robot-aided Bench-to-Clinics COVID-19 Neutralization Test workflow for the assessment of neutralization titers against SARS-CoV-2 variants within 24 hours after sample receiving.
The presented approach provides a valuable avenue for large-scale testing of clinical samples against SARS-CoV-2 and VOCs at BSL-2, supporting pandemic preparedness and response strategies.
- General concept comments
Here are some considerations/suggestions for the study:
1. Line 84, you mentioned that mGFP Δ3678 is more genetically stable than original mNG Δ3678. Is mNG Δ3678 genetically unstable? Is there any mutation during the passage of mGFP Δ3678? Any sequencing confirmation?
2. In the section of 2.2. Serum/plasma specimens, is there any demographic data available for these samples?
3. Why use VeroE6-TMPRSS2 cells rather than VeroE6 cells or VeroE6-hACE2 cells in the construction of mGFP Δ3678 and Omicron subvariants BA.5 and XBB.1.5? What kind of TMPRSS2 is expressed on the cell surface, human or monkey?
- Specific comments:
1) Line 138, typo of millimeter.
2) Line 147, Vero E6 cells or VeroE6-TMPRSS2 cells here?
3) Line 166 and Figure 1D, while gel analysis can confirm the PCR product size, it cannot tell whether the amplified gene is mGFP or not, or is there any mutation inside the amplified region.
4) Line 179, what samples are used here?
5) Line 196, typo of (FFRNT) here.
6) Line 284, quotation error of “16” here? Besides, Figure 1E should be mentioned here.
7) Line 326, (H) should be (F).
8) Line 430, (Figure 1D) should be (Figure 1F).

Fine.
Author Response
Major comments:
In this study, Jing Zou et al. presented an integrated research clinical platform for live SARS-CoV-2 neutralization assay, utilizing a highly attenuated SARS-CoV-2 (mGFP Δ3678_WA1/BA.5/XBB.1.5-spike). The viruses exhibited robust fluorescent signals in infected cells, and neutralization titers in an optimized fluorescence reduction neutralization assay that highly correlates with the conventional plaque reduction assay.
Furthermore, they established a streamlined robot-aided Bench-to-Clinics COVID-19 Neutralization Test workflow for the assessment of neutralization titers against SARS-CoV-2 variants within 24 hours after sample receiving.
The presented approach provides a valuable avenue for large-scale testing of clinical samples against SARS-CoV-2 and VOCs at BSL-2, supporting pandemic preparedness and response strategies.
Response: We thank the reviewer’s positive comments.
- General concept comments
Here are some considerations/suggestions for the study:
- Line 84, you mentioned that mGFP Δ3678 is more genetically stable than original mNG Δ3678. Is mNG Δ3678 genetically unstable? Is there any mutation during the passage of mGFP Δ3678? Any sequencing confirmation?
Response: We thank the reviewer’s comment. Both mGFP Δ3678 and mNG Δ3678 are genetically stable. The improved properties meant strong fluorescence signals in infected cells and neutralization assay sensitivity. To clarify this point, we have made these changes in the text accordingly.
No consensus mutations have been found by next-generation sequencing. We had also included this point in the manuscript.
- In the section 2.2. Serum/plasma specimens, is there any demographic data available for these samples?
Response: We thank the reviewer’s comment. Unfortunately, the demographic data is not available because we are not able to access patient’s information based on the current IRB protocol. In addition, we believe the demographic data is not relevant to the scope of the current study.
- Why use VeroE6-TMPRSS2 cells rather than VeroE6 cells or VeroE6-hACE2 cells in the construction of mGFP Δ3678 and Omicron subvariants BA.5 and XBB.1.5? What kind of TMPRSS2 is expressed on the cell surface, human or monkey?
Response: We thank the reviewer’s question. VeroE6-TMPRSS2 cells that expressed human transmembrane serine protease 2 (TMPRSS2) were used in this study, due to their high susceptibility to SARS-CoV-2 infection (referred to doi: 10.1073/pnas.2002589117). In addition, based on our experience, compared with VeroE6, VeroE6-TMPRSS2 has a lower chance of inducing deletion of furin-like cleavage site from the spike glycoprotein. Overexpressed hACE2 in VeroE6-hACE2 can interfere with viral infectivities during viral stock production. Collectively, VeroE6-TMPRSS2 cells instead of VeroE6 and VeroE6-hACE2 cells were used to maintain the genetic stability and high infectivities of stock viruses. We have added additional information in the revised manuscript.
- Specific comments:
1) Line 138, typo of millimeter.
Response: Corrected.
2) Line 147, Vero E6 cells or VeroE6-TMPRSS2 cells here?
Response: Corrected. VeroE6-TMPRSS2 cells were used here.
3) Line 166 and Figure 1D, while gel analysis can confirm the PCR product size, it cannot tell whether the amplified gene is mGFP or not, or is there any mutation inside the amplified region.
Response: Besides the gel analysis, next-generation sequencing was used to verify the sequence of passaged viruses. No consensus changes had been found in the passaged viruses.
4) Line 179, what samples are used here?
Response: Corrected.
5) Line 196, typo of (FFRNT) here.
Response: Corrected.
6) Line 284, quotation error of “16” here? Besides, Figure 1E should be mentioned here.
Response: Corrected.
7) Line 326, (H) should be (F).
Response: Corrected.
8) Line 430, (Figure 1D) should be (Figure 1F).
Response: Corrected.
Reviewer 2 Report
The manuscript is well written with appropriate descriptions of the methodology and presentation of results. My issue is that the authors don't make it clear in the introduction that there is already a neutralisation test using the same virus but with a neon green reporter (mNG) gene. The authors do quote this paper (reference 16) in the materials and methods but only in the context of using the delta 3678 backbone virus. They do not mention in the introduction that a fluorescent reporter neutralisation test was already developed by the original researchers, who developed the backbone virus. They do allude to the mNG reporter virus in the discussion, line 422 and say they have shown superiority of the mGFP assay to the mNG one. Where is the evidence for that in the results? Was a direct comparison carried out? They also say that mNG has been shown to be brighter in vitro and in vivo which diminishes the case for using mGFP. The FRNT assay may be a superior assay that they have developed but this is probably not dependent on the choice of fluorescent reporter gene. Overall, the authors need to spell out better what previous work they are building on and explain that the main advancement over previous work is the development of the FRNT assay.
There are some minor points:
1. Line 50. By 'functional' antibody do they mean 'neutralising'. All antibodies have function so please change the term.
2. Line 199. How many PFU were in the equal volume of serum?
3. Line 138 should be millilitre (not millimeter)
English is generally Ok. 'The' and 'a' left out in a number of sentences.
Author Response
Major point:
The manuscript is well written with appropriate descriptions of the methodology and presentation of results. My issue is that the authors don't make it clear in the introduction that there is already a neutralisation test using the same virus but with a neon green reporter (mNG) gene. The authors do quote this paper (reference 16) in the materials and methods but only in the context of using the delta 3678 backbone virus. They do not mention in the introduction that a fluorescent reporter neutralisation test was already developed by the original researchers, who developed the backbone virus. They do allude to the mNG reporter virus in the discussion, line 422 and say they have shown superiority of the mGFP assay to the mNG one. Where is the evidence for that in the results? Was a direct comparison carried out? They also say that mNG has been shown to be brighter in vitro and in vivo which diminishes the case for using mGFP. The FRNT assay may be a superior assay that they have developed but this is probably not dependent on the choice of fluorescent reporter gene. Overall, the authors need to spell out better what previous work they are building on and explain that the main advancement over previous work is the development of the FRNT assay.
Response: We thank the reviewer’s comments. We briefly described the mNG Δ3678 and mNG Δ3678-derived neutralization assay in the original manuscript (lines 76-77). In the revision, we have included additional clarifications on the limitation of the mNG Δ3678-derived neutralization assay. The superiority of mGFP over mNG can be found via two evidences: i) mGFP Δ3678 resulted in larger and brighter foci than mNG Δ3678 (Figure 1E & S2); ii) mGFP Δ3678 (R2=0.78) exhibited a higher correlation coefficient than mNG Δ3678 (R2=0.65). These comparisons were performed at the same time within the same set of human serum samples, although we published the mNG Δ3678 related data earlier. The other groups showed mNG is brighter in vitro and in vivo than GFP which does not contain the 76 nucleotide acid changes. Therefore, our data strongly suggest that the proper design of reporter sequences may enhance assay robustness. We have further clarified these points in the Introduction and discussion.
There are some minor points:
- Line 50. By 'functional' antibody do they mean 'neutralising'. All antibodies have function so please change the term.
Response: Corrected.
- Line 199. How many PFU were in the equal volume of serum?
Response: We used 100 FFU. The change has been made in the text.
- Line 138 should be millilitre (not millimeter)
Response: Corrected.
Reviewer 3 Report
This is a very well written and interesting paper. I am sure it will be well-received.
Line 63 – blockage or inhibition?
Line 117 – medium – it appears only one basic medium was used.
Line 273 – I am not sure this discussion should be in the M&M. It is not necessary to explain or justify the continuation to 3.2 and 3.3.
Line 309 – careful with the word “showed” – just use a simpler phrase like “There was no detectable neutralizing antibody in samples …..”
Line 320 – this is a very important and very good figure – I would suggest that the figure be enlarged.
Line 430 – please review sentence structure – Just state the observations. It is not necessary of helpful to say: “This study showed that…”.
Line 444 – very well constructed paragraph.
NA - minor editing required
Author Response
Major comments:
This is a very well written and interesting paper. I am sure it will be well-received.
Response: We thank the reviewer’s positive comments.
Line 63 – blockage or inhibition?
Response: Corrected.
Line 117 – medium – it appears only one basic medium was used.
Response: Corrected.
Line 273 – I am not sure this discussion should be in the M&M. It is not necessary to explain or justify the continuation to 3.2 and 3.3.
Response: We would like to keep the statement to better explain the difference between our current and previous studies.
Line 309 – careful with the word “showed” – just use a simpler phrase like “There was no detectable neutralizing antibody in samples …..”
Response: Corrected.
Line 320 – this is a very important and very good figure – I would suggest that the figure be enlarged.
Response: Corrected.
Line 430 – please review sentence structure – Just state the observations. It is not necessary of helpful to say: “This study showed that…”.
Response: Corrected.
Line 444 – very well constructed paragraph.
Response: We thank the reviewer’s positive comments.
Reviewer 4 Report
Review of MS#2583514 for Viruses
This manuscript describes the development a SARS-Cov2 neutralisation assay for testing of clinical samples in BSL-2 work environment. This is important since other live virus assays are generally regulated as BSL-3 and thus of restricted use since such facilities are not generally available in clinical testing laboratories.
The authors have accomplished this by developing an attenuated version of SARS-Cov2 virus, incorporating a reporter gene (mGFP). This construct has been conditionally approved for use at BSL-2 by the NIH.
The authors use this backbone to test reporter genes and Omicron subvariant virus types in development of a fluorescence reduction neutralisation assay (FRNT) for testing on patient serological samples.
They further show that this assay is suitable for incorporation into an automated platform for the high throughput processing of clinical samples.
In my opinion this could provide a very useful contribution to the clinical screening for SARS-CoV2 infections in the community. General implementation will require the full approval of the attenuated virus constructs for use at BSL-2, this it acknowledged by the authors.
The data presented is through and convincing.
I consider that a little more information on the automation platform used would be advantageous. Especially what turn around could be achieved. The authors state that samples can be processed within 24hrs but do not give an indication on how many samples can be processed in that time and what equipment might be needed. Also, an indication of cost would be very useful.
Overall I think this is an impressive piece of work, well worthy of publication in viruses.

Author Response
Major comments:
This manuscript describes the development a SARS-Cov2 neutralisation assay for testing of clinical samples in BSL-2 work environment. This is important since other live virus assays are generally regulated as BSL-3 and thus of restricted use since such facilities are not generally available in clinical testing laboratories.The authors have accomplished this by developing an attenuated version of SARS-Cov2 virus, incorporating a reporter gene (mGFP). This construct has been conditionally approved for use at BSL-2 by the NIH. The authors use this backbone to test reporter genes and Omicron subvariant virus types in development of a fluorescence reduction neutralisation assay (FRNT) for testing on patient serological samples. They further show that this assay is suitable for incorporation into an automated platform for the high throughput processing of clinical samples.
In my opinion this could provide a very useful contribution to the clinical screening for SARS-CoV2 infections in the community. General implementation will require the full approval of the attenuated virus constructs for use at BSL-2, this it acknowledged by the authors. The data presented is through and convincing.
I consider that a little more information on the automation platform used would be advantageous. Especially what turn around could be achieved. The authors state that samples can be processed within 24hrs but do not give an indication on how many samples can be processed in that time and what equipment might be needed. Also, an indication of cost would be very useful.
Overall I think this is an impressive piece of work, well worthy of publication in viruses.
Response: We thank the reviewer’s positive comments.
The throughput of the system and cost per sample can be varied. In this study, we proposed a prototype FRNT system. This system utilized a 96-well microplate format and the viruses that were stored at room temperature. Considering the limited assay time of cells (reasonable cell density for infection and readout) and virus (half-life of virus is around 3 hours at room temperature), the current assay was performed in batches (every 3 hours per batch). For each batch, the current system can process around 144 serum samples. The overall throughput can be further improved to more than 1000 samples per 8-hour period by preserving the virus at a cold temperature and performing the assay in a 384-well plate format.
The cost estimation will consider sample amounts, consumables, personnel, instrumentation, and facility maintenance. Given the high throughput of this prototype FRNT at BSL-2, the cost per sample can be less than traditional PRNT at BSL-3. In addition, the assay cost per sample can be significantly reduced when throughput increases. We have included this point in the Discussion.
Round 2
Reviewer 1 Report
I think that the manuscript has been improved, and the authors have addressed most of my concerns.

Fine.